# Alterations of the interactome of Bcl-2 proteins in breast cancer at the transcriptional, mutational and structural level

Simon Mathis König[1], Vendela Rissler[1], Thilde Terkelsen[1], Matteo Lambrughi[1], Elena Papaleo[1,2]*

**1** Computational Biology Laboratory, Danish Cancer Society Research Center, Copenhagen, Denmark, **2** Translational Disease Systems Biology, Faculty of Health and Medical Sciences, Novo Nordisk Foundation Center for Protein Research University of Copenhagen, Copenhagen, Denmark

* elenap@cancer.dk

**Data Availability Statement:** The data and scripts are deposited in a Github repository: https://github.com/ELELAB/bcl_bh3only_breast_cancer.

## Abstract

Apoptosis is an essential defensive mechanism against tumorigenesis. Proteins of the B-cell lymphoma-2 (Bcl-2) family regulate programmed cell death by the mitochondrial apoptosis pathway. In response to intracellular stress, the apoptotic balance is governed by interactions of three distinct subgroups of proteins; the activator/sensitizer BH3 (Bcl-2 homology 3)-only proteins, the pro-survival, and the pro-apoptotic executioner proteins. Changes in expression levels, stability, and functional impairment of pro-survival proteins can lead to an imbalance in tissue homeostasis. Their overexpression or hyperactivation can result in oncogenic effects. Pro-survival Bcl-2 family members carry out their function by binding the BH3 short linear motif of pro-apoptotic proteins in a modular way, creating a complex network of protein-protein interactions. Their dysfunction enables cancer cells to evade cell death. The critical role of Bcl-2 proteins in homeostasis and tumorigenesis, coupled with mounting insight in their structural properties, make them therapeutic targets of interest. A better understanding of gene expression, mutational profile, and molecular mechanisms of pro-survival Bcl-2 proteins in different cancer types, could help to clarify their role in cancer development and may guide advancement in drug discovery. Here, we shed light on the pro-survival Bcl-2 proteins in breast cancer using different bioinformatic approaches, linking -omics with structural data. We analyzed the changes in the expression of the Bcl-2 proteins and their BH3-containing interactors in breast cancer samples. We then studied, at the structural level, a selection of interactions, accounting for effects induced by mutations found in the breast cancer samples. We find two complexes between the up-regulated Bcl2A1 and two down-regulated BH3-only candidates (i.e., Hrk and Nr4a1) as targets associated with reduced apoptosis in breast cancer samples for future experimental validation. Furthermore, we predict L99R, M75R as damaging mutations altering protein stability, and Y120C as a possible allosteric mutation from an exposed surface to the BH3-binding site.

**Funding:** EP's group has been supported by LEO foundation grant (LF17006); Carlsberg Foundation Distinguished Fellowship (CF18-0314); Danmarks Grundforskningsfond (DNRF125); DECI-PRACE 14th HPC Grant CancerBH3; ISCRA-CINECA HP10C8YXRK and HP10CBLBWO. The funders had no role in study design, data collection and analysis, decision to publish, or preparation of the manuscript.

**Competing interests:** The authors have declared that no competing interests exist.

## Author summary

Apoptosis is a form of "cellular suicide". When the process of apoptosis is disrupted, cells that should have been eliminated may survive. The cellular decision-making between cell survival and cell death is orchestrated by interactions between three subgroups of Bcl-2 proteins. One of these, the pro-survival Bcl-2 subgroup, can enable cancer cells to escape cell death with impact on cancer development. Because of their role in cancers, pro-survival proteins are promising therapeutic targets for anti-cancer treatments. Despite progress in drug development, a common trait is that the available compounds could suffer in selectively targeting certain pro-survival proteins. We believe that additional knowledge on the alterations of the different pro-survival proteins in different cancer types, coupled with a better understanding of their interactions and functionality, will provide foundations for the development of new drugs with increased specificity. Here, we shed light on the role of the pro-survival proteins in breast cancer, by bridging different bioinformatic methods, linking the analysis of changes in gene expression with the study of structural ensembles of proteins. Our results highlight the prospects of an integrative bioinformatics approach for a comprehensive view of pro-survival proteins and their interactions in a specific cancer type.

## Introduction

Apoptosis is a vital physiological process for embryogenesis, maintaining tissue homeostasis, discharging damaged, or infectious cells. Failures in apoptosis may lead to carcinogenesis by favoring cell proliferation over cell death [1].

Apoptosis progresses through two discrete pathways: (i) intrinsic apoptosis (also called mitochondrial or stress-induced apoptosis), triggered by intracellular stresses, including oncogenic stress and chemotherapeutic agents [2], and (ii) extrinsic apoptosis, triggered by external stimuli detected by "death receptors" [3]. The intrinsic apoptotic pathway is governed by protein members of the B-cell lymphoma-2 (Bcl-2) family, dictating the cellular decision making between cell survival or programmed cell death [4]. As a response to cellular stress, these proteins preserve the integrity of the cell or commits the cell to apoptosis by permeabilization of the outer mitochondrial membrane (OMM) and release of proteins from the intermembrane space into the cytoplasm [5,6]. Regulation of the progression towards apoptosis is directed by interactions on the OMM between three distinct subgroups of the Bcl-2 family: the activator/sensitizer BH3 (Bcl-2 homology 3)-only proteins, the pro-survival inhibitor proteins, and the pro-apoptotic executioner proteins [7,8]. Proteins of the Bcl-2 family share amino acid sequences of homology known as Bcl-2 homology (BH) motifs.

The pro-survival proteins (Bcl-2, Bcl-xL/Bcl2l1, Bcl-w/Bcl2l2, Mcl-1, Bcl2a1/Bfl1, and Bcl2l10), along with the pro-apoptotic proteins (Bax, Bok, and Bak) share four BH motifs (BH1-4) [7,8]. They adopt a similar globular structure composed of nine α-helices, folding into a bundle, enclosing a central hydrophobic α-helix. This fold fosters a hydrophobic surface cleft, which constitutes an interface for the binding with BH3 motifs in other Bcl-2 family members. The globular multi-BH motif members of the Bcl-2 family mainly exert their apoptotic involvement at the OMM, to which they anchor by a C-terminal transmembrane region, exposing their globular helical bundle to the cytoplasm [9]. Unlike the globular Bcl-2 proteins, BH3-only proteins contain only one BH motif (BH3), which is often located in intrinsically disordered regions [10,11]. Like many other intrinsically disordered proteins, BH3-only proteins fold upon binding, by which their BH3 region becomes an amphipathic helix [11,12].

The BH3-only proteins can be divided into activators and sensitizers according to how they exploit their pro-apoptotic function [13,14]. Activators carry out their function by binding to pro-apoptotic proteins allowing the permeabilization of OMM and subsequent apoptosis event. Sensitizers bind to pro-survival members, inhibiting their binding with activator BH3 proteins and making them available to bind pro-apoptotic proteins.

Despite the importance of the BH3 motif in cell death regulation, a clear-cut definition of the motif is missing [15–17]. BH3-only proteins also feature distinct binding profiles and specificity toward Bcl-2 family members [18]. Attempts to define a consensus motif have returned motifs that are too strict (i.e., excluding proteins that are experimentally proved to bind Bcl-2 members) or too inclusive (i.e., reporting false-positives) [16]. A common feature, at the structural level, seems to be the presence of an amphipathic helix composing the BH3 motif that binds to the hydrophobic cleft on globular Bcl-2 members. This mainly happens by the insertion of four hydrophobic residues into hydrophobic pockets in the cleft and an invariant salt bridge between a conserved arginine residue in the Bcl-2 protein and a conserved aspartate in the BH3-only protein [19]. One of the four hydrophobic residues, an invariant leucine, packs against and form interactions with conserved residues in the hydrophobic cleft of the globular Bcl-2 proteins [19].

One of the cancer hallmarks is the capability to escape programmed cell death, for example, due to overexpression of pro-survival proteins [20–22]. Overexpression of pro-survival proteins is thought to contribute to tumorigenesis, the resistance of tumors to cytotoxic anticancer treatments, and increased migratory and invasive potentials [23–25].

Due to their pivotal role as inhibitors of apoptosis, pro-survival Bcl-2 proteins have been amenable therapeutic targets for drug discovery. Advances in the knowledge of their interactions with BH3-only proteins at the structural level have led to the development of inhibitors (BH3-mimetics), targeting the hydrophobic cleft of Bcl-2 proteins [26]. Some of these molecules suffer from issues related to specificity and selectivity and are often limited to specific cancer (sub)types [7]. To better exploit BH3-mimetics in cancer therapy, it is essential to elucidate the transcriptomic and mutational signature of the pro-survival proteins, as well as their interactions with BH3-only modulators in different cancer types. For example, several studies have demonstrated how the affinity of pro-survival proteins to these mimetics varies to a large extent. Increased levels of Bcl-2 promote sensitivity to ABT-263, whereas increased levels of Bcl-xL or Bcl-w conferred resistance to the same mimetic [27]. Similarly, two independent studies [28,29] found that cells overexpressing Bcl-2 showed high sensitivity to the BH3-mimetic ABT-737. In contrast, overexpression of Mcl-1 and Bcl2a1 in cell lines conferred resistance to a number of mimetics [29]. Comprehensive studies into the abundance, modifications, and interactome of each of the pro-survival Bcl-2 members for each cancer type would generate important knowledge, useful for optimization and design of BH3-mimetics. A system-biology approach is needed to achieve this goal, and the integration of different layers of bioinformatics tools could be beneficial.

Cancer develops when somatic mutations within the DNA alter specific amino acids in the protein-product, conferring selective advantages to highly proliferating cells [30]. Resistance to apoptosis is one of these selective advantages. Proteins are marginally stable under physiological conditions, and the substitution of single amino acids can alter their stability [31]. Besides protein stability, mutations can affect the binding affinity to interactors when occurring at, or near binding sites [32,33] or at distal sites, through complex allosteric mechanisms [34,35]. The quantitative analysis of the effects of mutations on both the stability and binding affinity is crucial for understanding the functional impact on pro-survival proteins. Another important aspect to consider is that mutations of Bcl-2 proteins might alter their sensitivity to BH3-mimetics [13,36]. For example, Fresquet et al. [37] found that in BCL-2-expressing

mouse lymphoma cells, two missense mutations within the BCL-2 gene conferred resistance to the BH3-mimetic ABT-199. Despite the importance of pro-survival Bcl-2 proteins, no comprehensive studies have been aimed at understanding their molecular mechanisms in cancer, through the investigation of the interface between the transcriptome, mutational signatures and the structural and functional effects of these alterations. Some of these aspects have only been analyzed individually and on specific case studies [21,38,39]. Here, we propose an integration of approaches to shed light on the pro-survival Bcl-2 proteins, by bridging two of the major branches of bioinformatics: (i) analysis of high-throughput sequencing data, and (ii) molecular modeling to unveil cancer-related alterations. We focused, as a case study, on Breast Invasive Carcinoma (BRCA) data from The Cancer Genome Atlas (TCGA) [40,41]. We identified a set of candidate genes encompassing the Bcl-2 family members and their protein interaction partners containing the BH3 motif, revising its definition according to recent findings [16]. We exploited differential expression analyses applied to RNA-Seq data from the TCGA-BRCA study to identify the expression levels of the candidate genes in tumor and normal tissues, and in different breast cancer subtypes. Additionally, we focused on the alterations in terms of missense mutations altering the protein product in the same cancer (sub)type. Next, we zoomed in at the structural level integrating different computational methods, which allowed for an assessment of the impact of mutations on protein function or stability. As a result, we provide a comprehensive picture of the most important protein-protein interactions within the Bcl-2 family and their alterations in breast cancer or breast cancer subtypes, which can be used as a guide for drug design or cancer target selection. Moreover, our study suggest new BH3-only proteins of interest within a breast cancer context that would be amenable for future experimental research.

## Results

### Definition of the BH3 motif

Our consensus motif was defined, comparing BH3 motifs previously described [16,42,43], along with structural information on important residues for interactions in canonical BH3-only proteins [17]. We defined a "loose" consensus motif to be permissive and avoid the loss of true-positives. A generalized motif composed of 10–13 residues was applied: [ar,h,s]-X (3,4)-L-X(2,3)-[ar,h,s]-[G,A,S,C]-X(0,1)-[D,E,Q,N]. *ar*, *h*, and *s* stand for aromatic (W, Y or F), aliphatic hydrophobic (V, I, L, M), and small residues (A, C, P), respectively (**Fig 1**). As a consequence, the positions 1, 3, and 5 of the motifs are the hydrophobic residues for the *h1*, *h2*, and *h3* hydrophobic pockets [7]. The leucine at position *h2* is very conserved [44], whereas the hydrophobic residues for the other binding pockets might vary in size and properties. The position 6, which was originally expected to require a glycine, can tolerate other residues with small side chains, such as alanine, serine, or cysteine [16]. The position 8, which was defined as an invariant aspartate, tolerates substitutions to glutamate, asparagine, or glutamine, as suggested by a recent study [16] and by the BH3-like motif of an autophagic protein [45,46].

### Identification of BH3-containing interaction partners and Bcl-2/BH3 interaction network

We retrieved the experimentally known Bcl-2 family interaction partners from the human *Integrated Interaction Database (IID)* [47]. We filtered the interaction list to retain only the proteins that included the BH3 motif. We collected 560 protein-protein interactions for the Bcl-2 family members (Bcl-2, Bcl-xL/Bcl2l1, Bcl-w/Bcl2l2, Mcl-1, Bcl2l10, Bcl2a1, Bok, Bax, Bak, Bcl2l12, Bcl2l13, Bcl2l14, and Bcl2l15, **S1 Table**) and 295 of them were selected as possible

**[ar,h,s]**-X(3,4)-**L**-X(2,3)-**[ar,h,s]**-*[G,A,S,C]*-X(0,1)-*[D,E,Q,N]*

*h1*　　　*h2*　　　*h3*

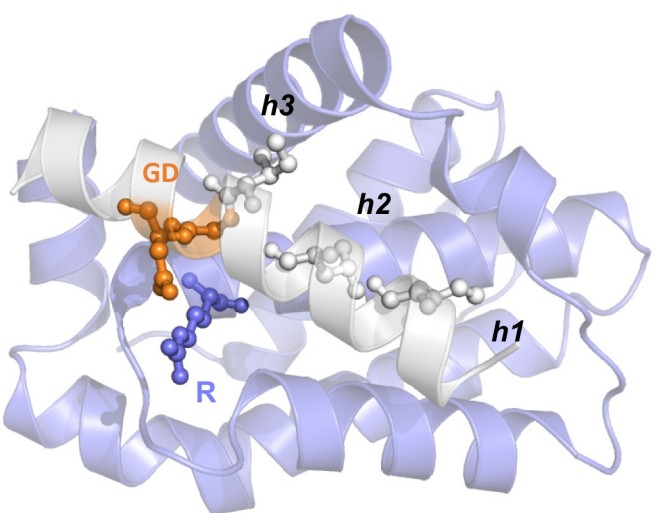

**Fig 1. BH3 motif and key positions for Bcl-2/BH3 interaction.** We illustrate the definition of the BH3 motif used in this study, highlighting the most important conserved residues for the binding to the BH3-binding groove of Bcl-2 family members. We used the complex of Bcl-xL and Bim as a reference (PDB entry 4QVF). We highlight the hydrophobic or aromatic residues which can occupy *h1* and *h2* hydrophobic pockets, along with the invariant leucine for h3 and the salt-bridge between the BH3 aspartate residue and the arginine of the Bcl-2 proteins.

BH3-containing proteins (**S2** and **S3** Tables, **Fig 2**). Among the 295 proteins, 282 can be classified as BH3-only (**S2** and **S3** Tables). The resulting protein-protein interaction network is compact and divided into only two connected components (**Fig 2**). The main connected component includes most of the Bcl-2 members and BH3-only proteins (291 nodes). An isolated small component refers to Bcl2l15 and its three BH3-only interactors (Meox2, Tead2, and Sdcbp). Most of the nodes feature a degree lower than ten, and collectively the average number of neighbors is 3.051. The most important hubs in the network (i.e., nodes connected to other nodes with a degree higher than the average connectivity in the network) are Bcl-2 (degree of 108), Bcl2l1 (96), Bax (88), Mcl-1 (62), Bak1 (33), Bcl2l2 (22), and Bcl2a1 (19) (**Fig 2**). Most of these proteins also correspond to the ones with high values of closeness centrality, which measure important nodes for the network communication (see GitHub repository).

We compared our predictions to a manual curation from literature of experimentally validated BH3-only targets, where we identified 26 of our candidates as known canonical or non-canonical BH3-containing proteins (Atg12, Aven, Beclin-1, Blid, Bnip1, Bnip2, Bop, Clu, Huwe1, Antxr1, Bbc3/Puma, Bcl2l11/Bim, Bad, Bid, Bik, Bmf, Hrk, Pmaip1/Noxa, Itm2b, Moap1, Rad9a, Spns1, Casp3, Pcna, Mycbp2 and Ambra1, **S2 Table**). The remaining predicted BH3-only targets will require further verification upon analyses of the corresponding three-dimensional (3D) structures to confirm that the motifs could fulfill the requirement for a BH3 motif (i.e., in disordered or solvent-exposed helical structures). Nevertheless, this list provides a rich source of information for experimental validation of new BH3-containing proteins.

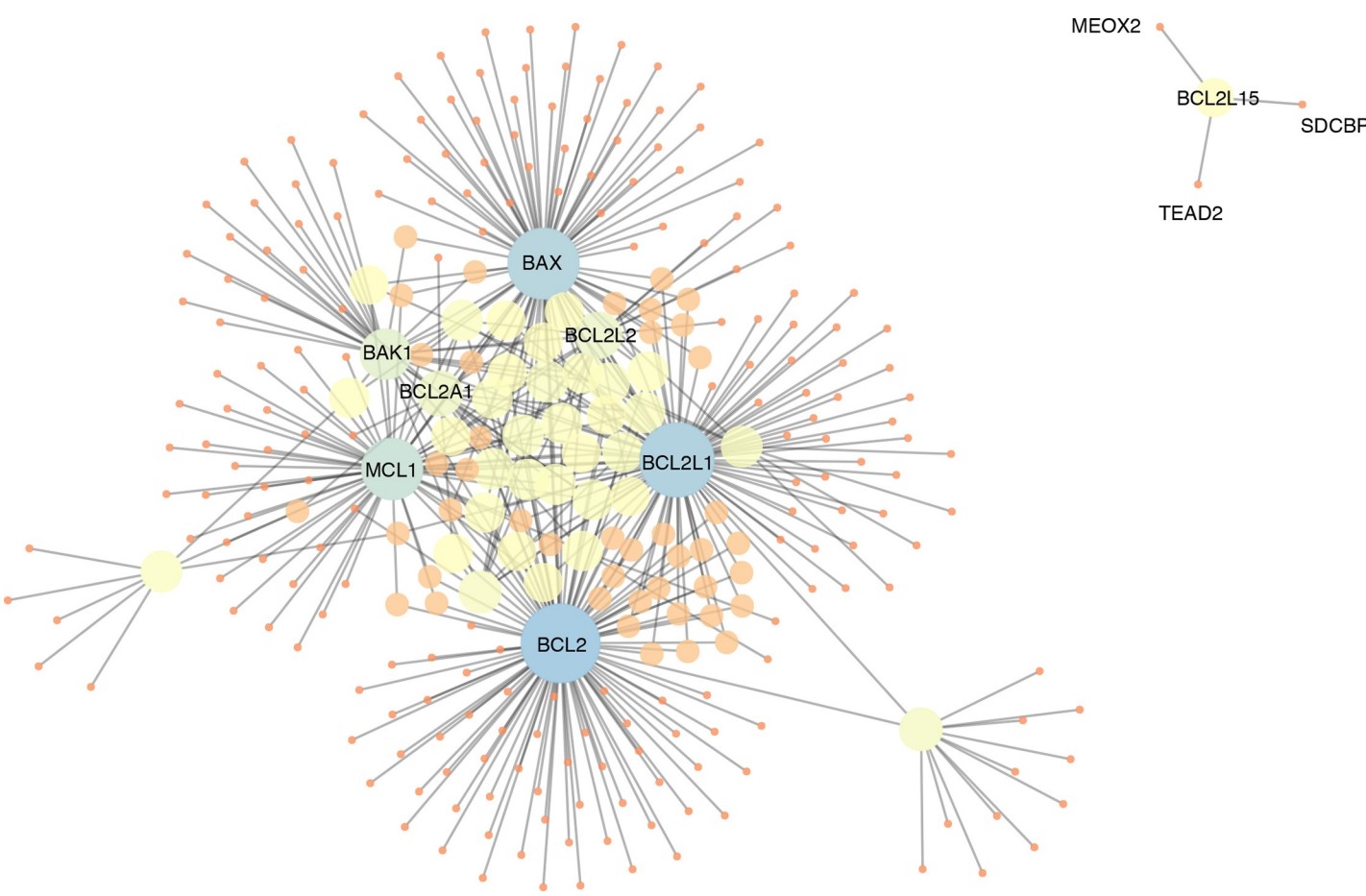

**Fig 2. Protein-protein interaction network of Bcl-2 proteins and BH3-containing interactors in *IID*.** We show the two connected components of the interaction network among the predicted BH3-only proteins and Bcl-2 pro-survival or pro-apoptotic globular proteins. The nodes of the network are depicted with different size and shade of colors as a function of the degree (i.e., the number of edges for each node). It results that the most important hubs are the pro-survival Bcl-2, Bcl2l1, Mcl-1, Bcl2l2 and Bcl2a1, along with the pro-apoptotic Bax and Bak1.

## Changes in gene expression of Bcl-2 family members in breast cancer

Firstly, we investigated the genes, encoding globular Bcl-2 members and their putative BH3-like interactors, characterized by changes in expression levels between tumor and tumor-adjacent normal tissues. The aggregated TCGA BRCA dataset contains 1102 tumor and 113 tumor-adjacent normal tissue samples. Breast cancer is a heterogeneous disease at both the morphological and molecular levels. To increase our understanding of biologically induced variation, we included PAM50 molecular subtype information, classifying breast carcinomas into subtypes based on variations in gene expression patterns. As a result, we excluded samples lacking subtype information. We performed differential expression analysis (DEA) on a final dataset, after pre-processing, normalization, and filtering, containing 14273 protein-coding genes and 444 tumor samples, including subtype information (Luminal A, Luminal B, Basal-like, and HER2-enriched) and 113 tumor-adjacent normal tissues samples. The Normal-like subtype was filtered out as it only encompassed four samples.

To quantify the magnitude and significance of differential expression between the conditions, tumor and normal samples, we employed *limma-voom* [48]. We find 3092 differentially expressed genes, of which 1738 down-regulated and 1354 up-regulated in tumor compared to

**Table 1. Differentially expressed BCL-2 genes in breast cancer subtypes.** The logFC is indicated in the table, all the results refer to an FDR < 0.05. Empty lines indicate that the gene is not differentially expressed in the corresponding comparison. BCL2A1 results the only pro-survival BCL-2 gene which is up-regulated in the majority of the comparisons. We notice that one of the main pro-survival genes, i.e. BCL-2 is mostly down-regulated in the TCGA-BRCA samples with HER2 and Basal subtypes. We do not report the comparison between LumA and LumB since none of the BCL-2 family members is deregulated in this comparison.

| GENE | Cancer vs Normal | Basal vs HER2 | Basal vs LumA | Basal vs LumB | Basal vs Normal | HER2 vs LumA | HER2 vs LumB | HER2 vs Normal | LumA vs Normal | LumB vs Normal |
|---|---|---|---|---|---|---|---|---|---|---|
| BCL-2 | | | -2.35 | -1.91 | -2.20 | -2.26 | -1.82 | -2.11 | | |
| BCL2A1 | 1.31 | 1.25 | 2.02 | 1.61 | 2.77 | | | 1.51 | | 1.15 |
| BCL2L1 | | | | | | | | | | |
| BCL2L2 | -1.01 | | | | -1.17 | | | -1.21 | | |
| BCL2L10 | | | | | | | | | | |
| MCL-1 | | | | | | | | | | |
| BAK1 | | | | | 1.34 | | | 1.03 | | |
| BAX | | | | | | | | | | |
| BOX | | | | | -1.46 | | | -1.47 | -1.06 | |

normal tissues. Among these genes, 45 candidate genes (Bcl-2 family members or their BH3-only interactors) are differentially expressed with 21 of them up-regulated and 24 down-regulated (S4 Table, Table 1, Fig 3). BCL2A1 is the only up-regulated pro-survival BCL-2 gene in the majority of the comparisons. We also noticed that one of the main pro-survival genes, i.e., BCL-2 is down-regulated in the TCGA-BRCA samples with HER2 and basal subtypes.

We then analyzed the patterns of deregulation in the BH3-containing candidate genes (S4 Table, Table 2). BH3-containing proteins are often induced transcriptionally by cytotoxic stresses, and they can either inhibit pro-survival Bcl-2 proteins or act through direct activation of pro-apoptotic Bak, Bax and Bok [7]. We were interested in patterns of opposite deregulation

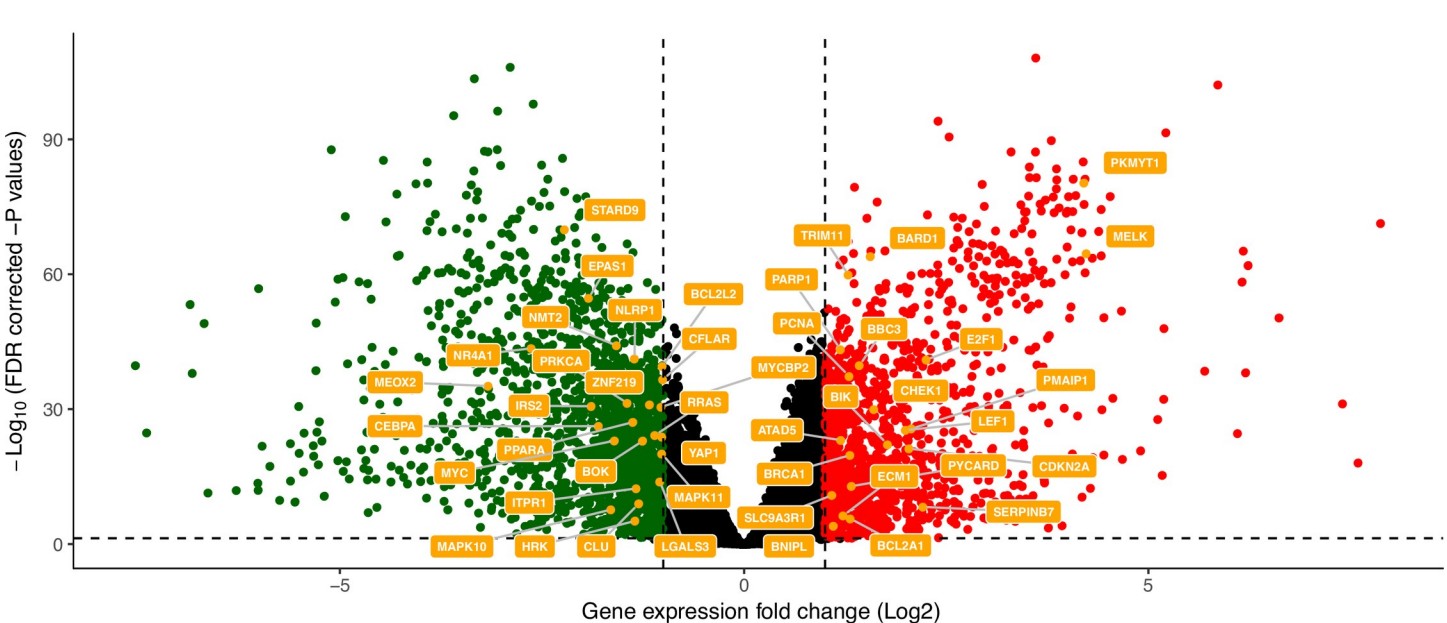

**Fig 3. Differentially expressed (DE) BCL-2 and BH3-containing genes.** We show as a volcano plot the BCL-2 and BH3-only genes, which are among the list of DE genes between tumor and normal tissues. The horizontal dashed line indicates the FDR cutoff (0.05). The vertical dashed lines represent the cutoffs in terms of logFc for up- and down-regulated genes. The other comparisons at the subtype level are reported in the GitHub repository associated to the publication and summarized in S4 Table.

**Table 2. Up-regulated BH3-only candidate genes in breast cancer subtypes.** The logFC is indicated in the table, all the results refer to an FDR < 0.05. Empty lines indicate that the gene is not differentially expressed in the corresponding comparison. We do not report the comparison between LumA and LumB since it results in only three BH3-only genes with signs of deregulation (see S4 Table). We show only BH3-containing genes for which we find differential expression in at least two comparisons, for sake of clarity. The full list is reported in S4 Table.

| GENE | Bcl-2 interactor | Cancer vs Normal | Basal vs HER2 | Basal vs LumA | Basal vs LumB | Basal vs Normal | HER2 vs LumA | HER2 vs LumB | HER2 vs Normal | LumA vs Normal | LumB vs Normal |
|---|---|---|---|---|---|---|---|---|---|---|---|
| RTN1 | Bcl2l1 | | | -1.65 | -1.13 | -2.12 | -1.44 | | -1.91 | | |
| NMT2 | Bcl-2 | -1.57 | 1.14 | 1.06 | 1.38 | | | | -1.82 | -1.74 | -2.06 |
| CLU | Bax, Bcl2l1 | -1.30 | | -1.15 | -1.02 | -2.17 | | | -1.47 | -1.01 | -1.15 |
| ZNF219 | Bcl2l1 | -1.16 | | | | -1.55 | | | -1.47 | | -1.21 |
| ECM1 | Mcl-1 | 1.22 | -1.72 | -1.86 | -1.85 | | | | 1.44 | 1.58 | 1.58 |
| SLC9A3R1 | Bcl2a1 | 1.08 | -1.42 | -1.48 | -2.17 | | | | 1.22 | 1.28 | 1.97 |
| HRK | Bcl-2, Bcl2l1, Bcl2l2, Mcl1, Bcl2a1, Bax | -1.34 | 2.22 | 3.79 | 3.38 | -1.33 | 1.57 | 1.16 | | -2.45 | -2.04 |
| IRS2 | Bcl-2, Bcl2l1 | -1.88 | 1.05 | | | -1.92 | -1.27 | -1.07 | -2.98 | -1.71 | -1.91 |
| NLRP1 | Bcl-2, Bcl2l1 | -1.35 | | | | -1.27 | | | -1.43 | -1.28 | -1.65 |
| LGALS3 | Bcl-2 | -1.04 | | | | -1.19 | | | | | -1.46 |
| RRAS | Bcl-2 | -1.10 | | | | -1.17 | | | | -1.01 | -1.39 |
| ITPR1 | Bcl-2 | -1.33 | -1.46 | -2.12 | -1.74 | -2.92 | | | -1.45 | | -1.17 |
| CFLAR | Bcl2l1, Bax | -1.00 | | | | | | | | -1.04 | -1.27 |
| NR4A1 | Bcl-2, Bcl2l10, Bcl2a1 | -2.63 | | | | -2.64 | | | -2.72 | -2.55 | -2.83 |
| STARD9 | Mcl-1 | -2.21 | | | | -2.10 | | | -2.50 | -2.16 | -2.43 |

of BH3-only candidates and pro-survival Bcl-2 family members, respectively. In particular, we expected an up-regulation of pro-survival Bcl-2 family members and a down-regulation of the cognate BH3-only genes in cancer samples. Other expression patterns of interest were the down-regulation of pro-apoptotic Bcl-2 members (BAX, BAK1 or BOK) and down-regulation of the cognate BH3-only genes. Only BOK is down-regulated, according to our analyses, whereas BOK-specific BH3-containing interactors are not deregulated. Consequently, we focused on the relationship between the pro-survival and BH3-containing genes.

Of note, the high overexpression of MELK (S4 Table) in most of the comparisons is likely due to effects unrelated to the apoptotic pathway, since this gene encodes an oncogenic kinase in breast cancer [49]. Similarly, the up-regulation of PCNA, CHECK1 or GZMB (S4 Table) might be related to other aspects of apoptosis or breast cancer pathways. PCNA is known as a marker for breast cancer [50], even if this should be verified at the protein level, considering that we analyzed only gene expression data. Associations with CHECK1 levels and breast cancer have also been reported [51].

A group of predicted BH3-only genes, found in the interactome of at least one Bcl-2 pro-survival protein, is highly down-regulated in all or most of the breast cancer subtypes compared to the normal samples. Hence, it would be interesting to assess if the corresponding protein products could bind and regulate other Bcl-2 family members, which are prone to up-regulation in breast cancer. BH3-only genes with these patterns are, for example, CLU, IRS2, NLRP1, NMT2, ITPR1, CFLAR, LGALS3, RRAS, RTN1, STARD9 and ZNF219. Our results, also in light of the intrinsic incompleteness of the annotations in protein-protein interaction databases, suggest that these genes could be interesting candidates for future studies, assessing their promiscuity of binding towards other Bcl-2 proteins, such as Bcl2a1.

SLC9A3R1, i.e., one of the Bcl2a1 interactors, is down-regulated in the Basal subtype, in parallel with up-regulation of the BCL2A1 gene in the same subtype (Table 1), suggesting an interesting association.

Of interest, we observe that the pro-survival BCL2A1 gene and its interactor HRK (Hara-kiri) are differentially expressed with up-regulation of the pro-survival protein and down-regulation of the inhibitor of apoptosis HRK in all the subtypes. Hrk is a promiscuous BH3-containing protein, considering that it has been found in the interactome of Bcl-2, Mcl-1, Bcl2l1, Bcl2l2 and, Bcl2a1 proteins (**Table 2**). In addition, our analyses of the Bcl-2/BH3 protein-protein interaction network pointed out a high closeness centrality score (0.48) for Hrk. Hrk has been experimentally showed to bind Bcl-2 and Bcl-xL [52]. Our results suggest that the interaction between Bcl2a1 and Hrk could be of interest to explore in breast cancer since their deregulation points in the direction of evading apoptosis, a cancer hallmark.

Nr4a1 might also be an interesting BH3-candidate for Bcl2a1 in breast cancer for similar reasons (**Table 2**). A Nr4a1-derived peptide has been reported with the capability to convert Bcl-2 into a pro-apoptotic molecule [53,54]. Our predictions suggest that Nr4a1 could include two potential BH3 motifs (at positions 201–213 and 386–398), which, if experimentally validated, could open new directions toward a multifaceted regulatory role of Nr4a1/Nur77 on Bcl-2 proteins. The two BH3 motifs of Nr4a1 are both placed in disordered, solvent-exposed regions, according to the analysis of the 3D structure of the C-terminal domain of the protein (PDB entry 4RZF, residues 351–598). These data are also in agreement with structural propensities from *FELLS* [55]. Therefore, they could be, in principle, accessible for interaction with the Bcl-2 family members.

## Models of interaction between Bcl2a1/Bfl1 and the BH3-only interactor Hrk

Along with changes in expression levels, other alterations that could trigger evasion of apoptosis in cancer can be related to somatic mutations in the coding region of the BCL-2 genes and their interactors. Such mutations exert an effect on the protein products of these genes. Before analyzing the TCGA-BRCA somatic missense mutations in the coding region of BCL2A1 and its interactors, we built a 3D structural model of their protein complexes.

Mutations can impact on a myriad of different aspects at the protein level, including protein stability or activity. Hence, it becomes fundamental to be able to assess them at different levels, as we recently showed in other works [56,57]. The knowledge of the structure of the targets of interest is important in this context. 3D structures of Bcl2a1 in complex with the BH3 regions of Bim, Bak, Noxa, tBid and Puma are available in the Protein Data Bank (PDB).

We thus employed comparative modeling to derive models of the 3D structure of the Bcl2a1 complex with the BH3-like sequences of Hrk, similarly to what we did for other short linear motifs in complex with folded proteins [58]. We collected two different models for the interaction of the two BH3-only motifs that we found in Hrk (i.e., 28–50 and 63–85). As a template for the modeling, we used the complex between Bcl2a1 and Puma (PDB entry 5UUL, [59]) as it is the Bcl2a1-BH3 complex with the best atomic resolution (1.33 Å) available. As a result of the comparative modeling approach, our target BH3 peptides are assumed to interact in a similar manner to the template structure at the binding interface. This assumption is supported by the conserved helical conformation of BH3 motifs upon binding to the Bcl-2 family members [7]. The comparative modeling approach used here is convenient to scrutinize the effect of mutations at the binding interface in a high-throughput manner. However, we acknowledge that this approach could suffer from limitation in the description of fine structural details and interactions that other more computationally-demanding and accurate methodologies could provide, such as sampling based on Montecarlo or Molecular Dynamics simulations.

**Fig 4A and 4B** shows the BH3 peptides of Hrk_1 (residues 28–50) compared to Puma (residues 132–154). The BH3 motif for Hrk_1, as found in our motif search, defines the first

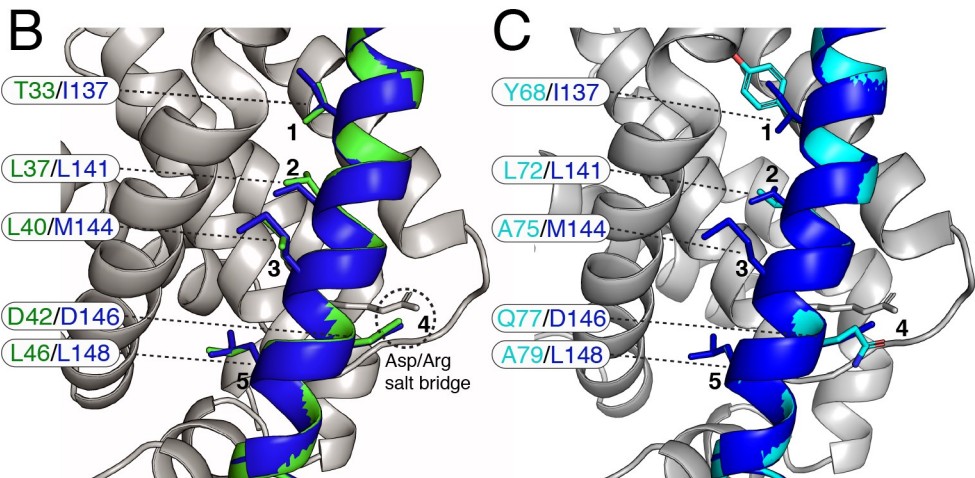

**Fig 4. The model of interaction of Bcl2a1 and Hrk BH3-like peptides.** A) The predicted BH3 regions of Hrk (i.e., Hrk_1, residues 28–50 and Hrk_2, residues 63–85) are aligned to the Puma BH3 as a reference. B-C) The 3D models of the complexes between Bcl2a1 and Hrk_1 or Hrk_2 are shown with the residues important for the intermolecular interactions, in comparison with the experimentally-derived complex.

residue as an aliphatic hydrophobic residue (L32 of Hrk_1). Nevertheless, Barrera-Vilarmau et al. [60] resolved a fragment of Hrk (residues 22–53, PDB entry 2L58) by NMR. They identified T33 opposed to L32 as the key residue in binding of Hrk with Bcl-2 and Bcl-xL. We thus modeled the complex Bcl2a1-Hrk_1 (28–50) aligning T33 as the binding residue in *h1*. The complex features hydrophobic residues at the positions 2,3 and 5 of the motif, with the invariant leucine in position 2 and the conserved aspartate occupying position 4 (**Fig 4B**).

A number of BH3-only proteins have been predicted to contain a transmembrane (TM) region, suggesting an OMM-anchoring function and in some cases, the BH3 itself can associate with membranes [61]. Barrera-Vilarmau et al. revealed a TM region in Hrk (residues 69–91) [60]. Given these results, we expect this second BH3-containing peptide (Hrk_2) to be part of the TM region and have a role in membrane targeting activities. Nevertheless, its association with membranes could be modulated by many factors and it may act as interactor for the Bcl-2 family members when it is not associated with the membrane. Likewise, in this case we find the hydrophobic residues at position 1, 2, 3, and 5. Position 4 is characterized by a glutamine (Q77) residue, which replaces the negatively charged aspartate (**Fig 4A and 4C**). Despite aspartate being a highly conserved key residue in the BH3 motif, other residues at this position have been reported in BH3-like motifs [16,46]. For example, the pro-autophagic protein Ambra1, which acts as an inhibitor of pro-survival Bcl-2 through a BH3-like motif, contains a glutamine residue instead of the conserved aspartate [46]. Another example is found in the BNIP group of proteins, where the aspartate is substituted by an asparagine [62]. Our structural analysis of Bcl2a1 /Hrk_2 (residues 63–85) complex, along with the other BH3-like sequences mentioned above, suggests that the invariant salt-bridge could be replaced by interactions between the

Bcl-2 arginine and residues such as asparagine or glutamine, which are still capable of providing a delocalized partial charge around their functional groups.

## Structure-based assessment of the effect of mutations in the Bcl2a1-BH3 complexes: Stability and local effects on interactions

We collected missense mutations for the proteins of interest in breast cancer, aggregating data from different cancer genomics projects (see Materials and Methods for details and S5 Table). Four mutations were reported for Bcl2a1 (M75R, L99R, Y120C and, V145L), whereas we did not identify missense mutations altering Hrk in the breast cancer samples under investigation.

We used the models of the two complexes between Bcl2a1 and the two Hrk BH3 regions, along with the structure of the complex between Bcl2a1 and Puma to predict the functional impact, in terms of binding free energies, of any possible substitutions of the Bcl2a1 protein and its BH3-containing interactors (Fig 5). We also used the same high-throughput pipeline to estimate the changes in free energies upon mutation associated to the structural stability of Bcl2a1, to be able to discriminate between effects that are related to its cellular function (i.e., the binding with the BH3-only proteins) or related to its stability, and thus likely to alter, for example, the protein turnover at the cellular level (see below).

In our mutational scans, $\Delta\Delta G$ values that are close to 0 kcal/mol indicates that the original residue is not essential for protein stability and/or complex formation. Negative $\Delta\Delta G$ values indicate that the mutant variant is more stable than the wild-type variant, whereas positive $\Delta\Delta G$ values indicates that the substitution has a destabilizing effect and that the wild-type amino acid may have an important function preserving the integrity of the protein structure or of the binding interface in the complex.

The high-throughput mutational approach allowed us to evaluate the effects, on Bcl2a1stability or its interaction with BH3-containing proteins, when substituting wild-type residues with the mutant variants found in cancer samples, allowing for a classification of potentially damaging and neutral cancer mutations. Moreover, it allowed us to evaluate the general effects of any amino-acid substitutions over the whole protein structure or complex, providing a useful set of precomputed $\Delta\Delta$Gs for future assessment or annotation of Bcl2a1 mutations. The latter could be a valuable source of information for future studies.

Additionally, the deep mutational scan increases our knowledge of hotspot residues for protein-peptide binding between Bcl2a1 and the putative BH3-like proteins. This knowledge could potentially aid in the design of selective peptide inhibitors targeting Bcl2a1.

We identified R88 of Bcl2a1, which is the arginine important for salt-bridge interactions with the conserved aspartate of the BH3 motif, as a sensitive hotspot for mutations (Fig 5B). This position does not tolerate any substitution, even to lysine. The only tolerated substitution is to histidine. The histidine side-chain has a pKa close to physiological pH, implying that the local amino acidic environment determined shifts in pH, will change its average charge. Consequently, we could expect a population of protonated positively charged histidine residues in the complexes at physiological conditions. Other critical positions in the BH3-binding cleft of Bcl2a1 are hydrophobic residues such as L52 (Fig 5A) and V74 (Fig 5B).

T33 of Hrk, the position earlier identified as one of the residues contributing to the binding in the BH3 motif of Hrk, is predicted to favor substitution to a hydrophobic amino acid in the form of either: one of the aliphatic amino acids (alanine, valine, leucine, and methionine) or the aromatic phenylalanine (Fig 5C). This result suggests that T in *h1* is suboptimal for binding to Bcl2a1 and this feature could be used to design a stronger binder. A34 and G41 of the BH3 peptide are predicted highly intolerant to any substitution except for a glycine and alanine, respectively. This suggests that the two BH3 positions adjacent to the residues for *h1* and

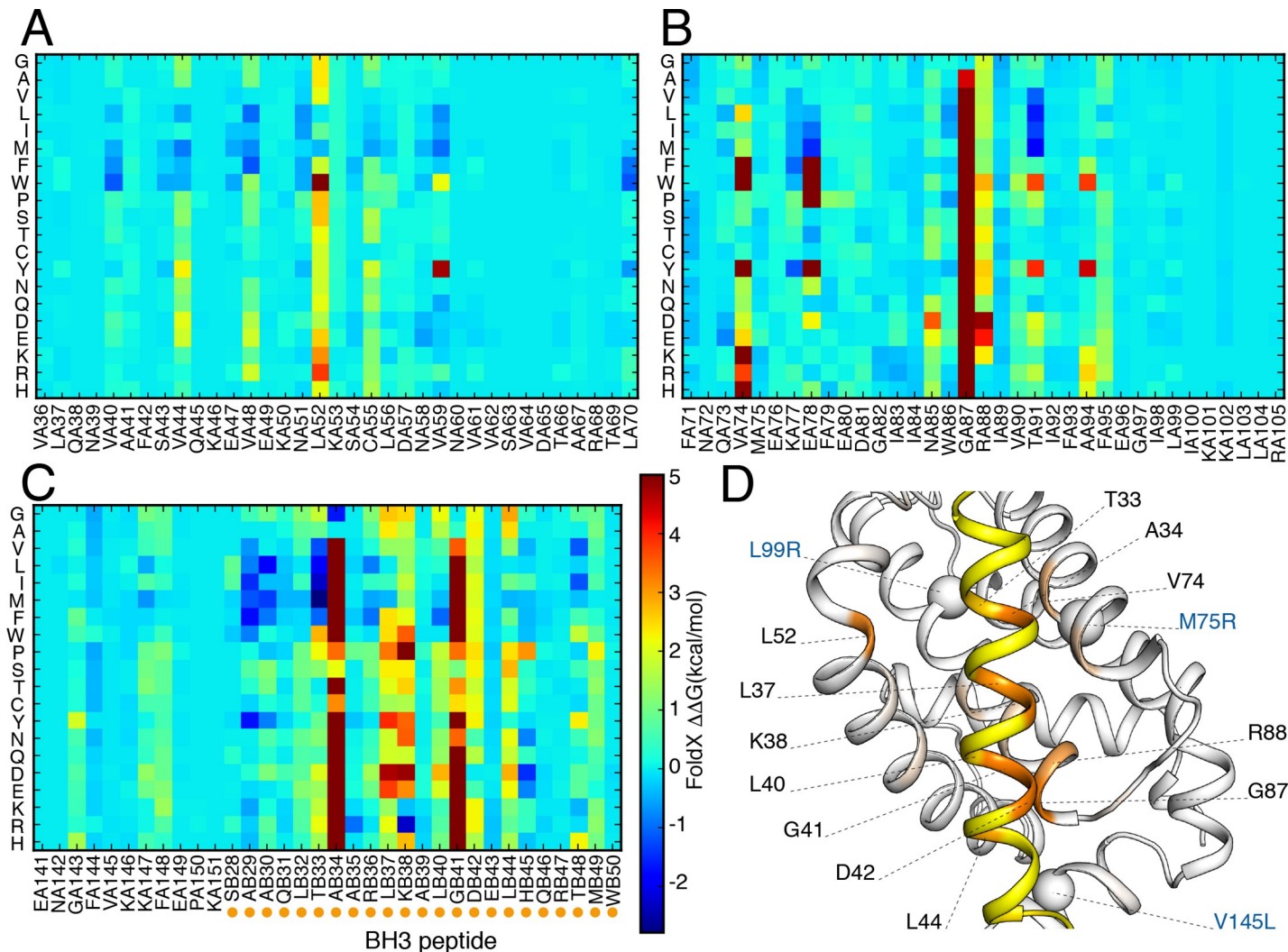

**Fig 5. *In silico* mutational scanning of the Bcl2a1-Hrk complexes using an empirical energy function.** The binding ΔΔGs are shown as heatmaps for the saturation mutagenesis carried out *in silico* on the Bcl2a1-Hrk_1 complex as a reference. In particular, we report the results for: A) α-helices 2 (residues 32–52), and 3 (residues 53–58) of Bcl2a1, B) α-helices 4 (66–80) and 5 (86–105) of Bcl2a1 and, C) Hrk_1 BH3 peptide (residue 28–50). The ΔΔGs have been truncated at a value of 5 kcal/mol for sake of clarity. D) The color gradient indicates the average ΔΔGs for mutations at each position in Bcl2a1 (white to red) and Hrk_1 (yellow to orange) on the 3D structure. The spheres indicate positions of three of the breast cancer mutations of Bcl2a1 (M75R, L99R and V145L). The data with the mutational scans of each complex are reported in the GitHub repository.

*h2* binding need to be of a small size and any other substitution would create a steric hindrance. We observe a similar trend in the mutational scan of the complex between Bcl2a1 and Puma (**S1 Fig**), providing important features for the definition of the BH3 motif.

The BH3 motif invariant leucine L37 in *h2* is predicted sensitive to most substitutions. L37 moderately tolerates substitutions by valine, isoleucine, and methionine. L40 (*h3*) and L44 (occupying a fourth hydrophobic pocket, *h4*) also contribute to the interaction interface and are predicted to tolerate substitutions only to aliphatic and aromatic amino acids.

The substitution of the conserved and salt-bridge forming aspartate (D42) is generally poorly tolerated. D42 can be replaced without any marked effects only by: (i) glutamate, which is also negatively charged (ii) asparagine and glutamine, which are similar in size to aspartate and glutamate but contain an amino group. This finding consolidates the notion that the selectivity at this BH3 site is triggered by the possibility of maintaining electrostatic-based

interactions with the arginine of the Bcl-2 protein. This may also occur in absence of negatively charged residues, if polar residues of a similar size are present, such as glutamine and asparagine. The original definition of the BH3 motif, which expected an invariant aspartate, should be revised to include this possibility so that a larger number of interactors could be identified.

In conclusion, the mutational scan of Bcl2a1-Hrk_1 suggests a subset of substitutions with damaging effects on the complex formations. These results highlight the importance of: i) hydrophobic residues in binding the amphipathic helix comprising the BH3 motif; ii) propensity to small side chains adjacent to the *h2* and *h3* hydrophobic residues; iii) the disadvantage of having a threonine (T33) instead of a hydrophobic amino acid in the *h1* pocket; and iv) the electrostatic interactions between arginine (R88) and the conserved aspartate (DB42), which can be fulfilled by other residues such as glutamate, glutamine and asparagine.

### Structure-based assessment of the effect of mutations in the Bcl2a1-BH3 complexes using Protein Structure Networks

The four reported cancer mutations in Bcl2a1, M75R, L99R, Y120C and, V145L are not located in the proximity of the BH3-binding domain and are not predicted to have any local effects on the complex formation (**Fig 5** and **S1 Fig**). M75 is in the proximity of the BH3 binding pocket, and it has been mentioned as an important residue in the hydrophobic pocket *h2* [63]. M75R is not predicted to alter binding ΔΔGs in our analyses, in agreement with the fact that most of the interaction with the BH3 peptide could be mediated by the backbone of this residue [64].

We investigated possible allosteric effects induced from these distal sites to the BH3 interface using a contact-based Protein Structure Network (PSN) approach [65,66]. Indeed, indirect and long-range effects of the mutations to the BH3-binding site cannot be captured by the high-throughput mutagenesis scan that we performed with the *FoldX* empirical energy function, as it is tailored to describe local rearrangements at the side-chain level only.

At first, we generated an ensemble of conformations (**Fig 6A**) around the structure of a reference Bcl2a1 -BH3 complex (see Materials and Methods for details) using a coarse-grained model and the *CABS-flex* sampling method [67,68]. The models were then reconstructed to all-atom representation before the PSN analysis. We used an ensemble of conformations to better model the inherent flexibility of the proteins in the complex, along with to remove spurious contacts from the PSN. We then analyzed the Bcl2a1 cancer mutation sites for their: i) propensity to act as hub residues in the network, i.e., nodes that features a high degree of connectivity and likely to be important for the maintenance of the protein architecture or for communication throughout the network; ii) propensity to communicate to the BH3-binding region, estimating the shortest paths of communication between each mutation site and a group of residues (V48, R88, L52, V74, T91, and F95, **Fig 6B**), which we selected as hotspots for binding by the deep mutational scan discussed above (**S1 Fig**).

M75 and L99 are hub residues in the PSN, whereas Y120 and, V145 do not show a hub behavior (**Fig 6C**). Y120 is solvent-exposed in the ensemble and, as such, unlikely to contribute with intramolecular interactions. V145, on the contrary, is partially buried in the protein core and next to a hub residue (F144).

As a complement to the hub analyses, we carried out the saturation mutational scan on the free-state of Bcl2a1 with the empirical energy function described above (**Fig 6D, S6 Table**). This allowed us to predict the impacts of the cancer mutations on the protein structural stability, which were in overall agreement with the hub results. Indeed, we predict M75R and L99R as damaging mutations for stability (ΔΔG of 3.32 and 6.33 kcal/mol, respectively). Y120C and V145L are predicted with neutral effects. L99 is also one of the Bcl2a1 hotspots upon the deep mutational scan (**Fig 6D**), suggesting the sensitivity of this site to mutations and, as a

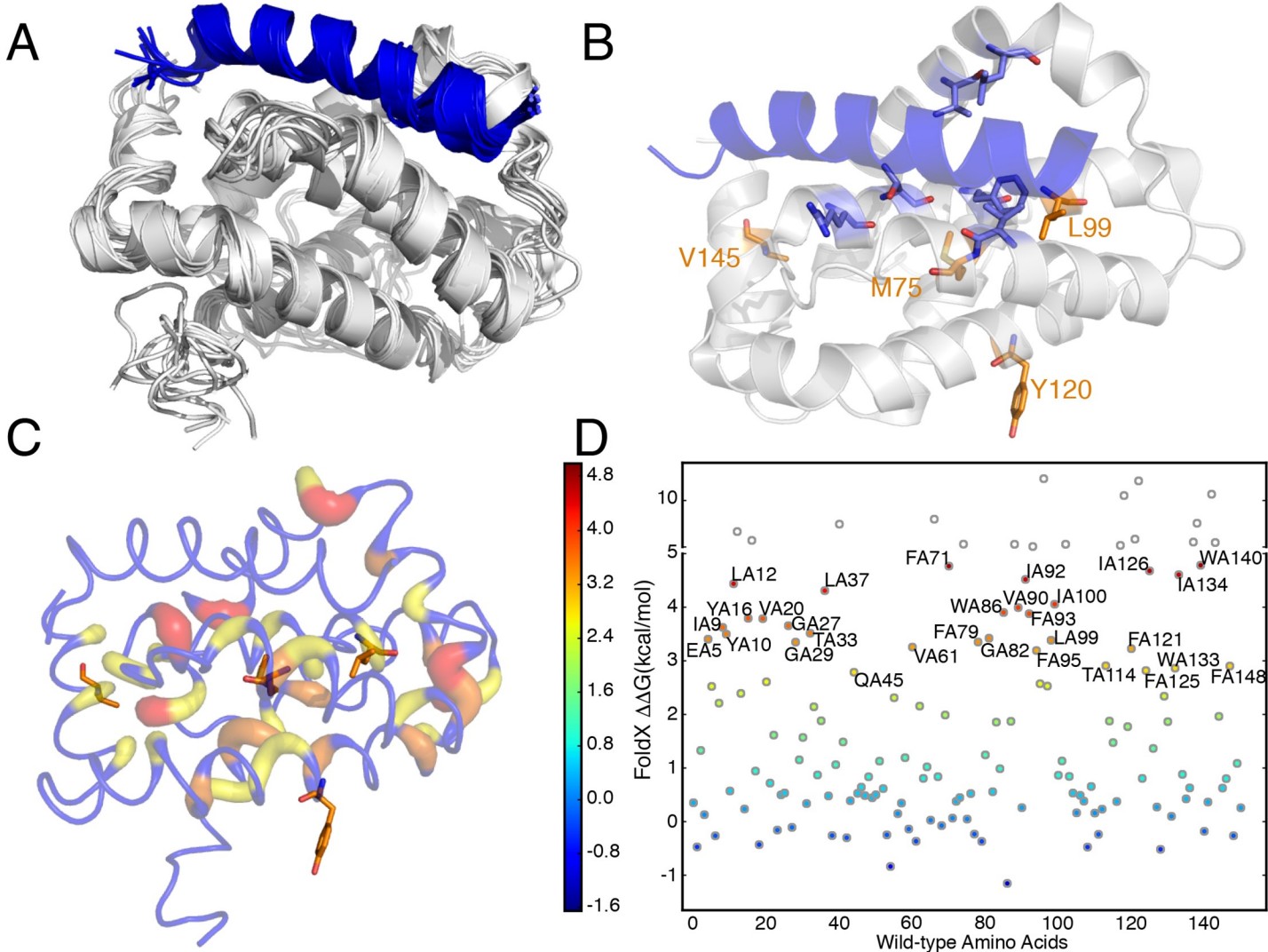

**Fig 6. Analysis of the mutation sites in light of PSN of a conformational ensemble of Bcl2a1-Puma complex. A)** The ten models of the conformational ensemble of the Bcl2a1-Puma complex generated by *CabsFlex 2.0* is shown. We used this ensemble of conformations for the PSN analysis. **B)** The Bcl2a1 cancer mutation sites and the target residues for path analyses (V48, R88, L52, V74, T91, and F95) are highlighted in orange and marine, respectively, using the X-ray structure of the complex between Bcl2a1 and Puma (PDB entry 5UUL) as a reference. **C)** The hub residues in the PSN of the Bcl2a1-Puma complex are shown with different scale of colors and cartoon thickness as a function of the degree (from yellow to red for degree from 3 to 5). The nodes that are not classified as hubs (degree < 3) are colored in blue. M75 and L99 are hub residues in the PSN, whereas Y120 and V145 do not show a hub behavior. **D)** Effects of amino acid substitutions on the free-state of Bcl2a1 upon *in silico* saturation mutagenesis to estimate ΔΔGs associated with protein structural stability. A scattered plot depicting the average ΔΔG of all the possible mutations in each position of the wild-type sequence of Bc2a1 is shown. The top 20 most destabilizing mutations are labeled. The labels follow the convention: residue type, chain ID, and residue number. Empty circles represent residues above the 5 kcal/mol cutoff, as explained in the Materials and Methods section. L99 is among the top 20 hotspots for protein stability, suggesting its sensitivity to any kind of mutations.

consequence, its importance for the Bc2a1 architecture. In support to these results, M75 and L99 are also highly conserved sites in 137 Bcl2a1 homolog protein sequences according to a *ConSurf* analysis (conservation scores of -0.952 and -0.761, respectively, **S7 Table**), whereas Y120 and V145 are poorly conserved.

Our analyses suggest that M75R and L99R mutations have the major effect of destabilizing the protein structure. This prediction could be validated assessing, for example, the cellular protein levels of the corresponding Bcl2a1 variants and their propensity for increased degradation by the proteasome, as recently shown for other cancer mutations [69,70].

**Table 3. Shortest paths of communication from Y120 to the BH3-binding interface of Bcl2a1.**

| Target interface residue | Path | Path length | Sum of weights | Average weight |
|---|---|---|---|---|
| VAL74 | Y120-124E-127M-9I-123A-13A-122V-71F-75M-74V | 10 | 480 | 53.3 |
| ARG88 | Y120-124E-127M-9I-126I-130T-84I-129N-79F-78E-88R | 11 | 510 | 51 |
| PHE95 | Y120-124E-127M-9I-123A-13A-122V-71F-75M-74V-F95 | 11 | 580 | 58 |

To assess the capability of the mutation sites to mediate long-range effects to the BH3 binding region, we calculated the shortest paths of communication between each mutation site and the interface residue probes depicted in **Fig 6B**. To reduce the risk of false-positive hits, we manually discarded paths that were not likely to act through a cascade of collisional events mediated by changes in residue side chains. This step included the removal of paths that involved the first neighbors of the mutation site in the sequence space, or topological paths that were related to secondary structures. Y120 turned out to be a possible residue that from an accessible surface could communicate to three of the interface residues selected as probes in the path analyses, i.e., R88, V74 and F95 (**Table 3**). In all the cases, the communication passes through a conserved group of nodes around Y120 (E124, M127, and I9).

Y120C mutation could also impair post-translational modifications or protein-protein interactions. According to literature searches, *PhosphoSite* [71] and predictions with *NetPhos* [72] Y120 is unlikely to be a post-translational modification site, suggesting that we could rule out the hypothesis of a disrupted post-translational modification.

In addition, we used a structure-based statistical mechanical model implemented in *AlloSigMA*[73,74] to obtain a direct estimate of the allosteric effects caused by the Y120C mutation. We used the crystallographic structure of Bcl2a1 in complex with Puma BH3 (PDB entry 5UUL). We also used a conformation of Bcl2a1 with Puma BH3 from the *CABS-flex* ensemble, in which Y120 and its surroundings showed conformational changes to account for the inherent flexibility in this region. In the mutation Y120C, a bulky large aromatic residue is substituted with a smaller one. Therefore, we expect that this substitution could cause a loosening in the network of protein contacts (i.e., we defined it as a DOWN mutation in *AlloSigMA*). The two structures give similar results, predicting local increased dynamics (i.e., a possible destabilization of the contact network) for the residues located in the α-helix 6 (residues T114-M127), and in the N-terminal α-helix 1, (residues F6-G7 and I9-Q14) (**Fig 7**). Among these residues, we identify some of the nodes mediating the communication from Y120, such as E124, M127 and, I9. Furthermore, we observe a distal effect on R68 and E96 sites (**Fig 7**). We observe that these residues are localized in the proximity of the ones that we selected as probes for the PSN-based path, i.e., F95 and V74 (Fig 6 and Table 3). Overall, our data suggest that the mutation Y120C has a destabilizing impact on the network of contacts in Bcl2a1 and can affect its allosteric communication to the residues in the proximity of the BH3-binding interface.

## Discussion

The network of protein-protein interactions between globular Bcl-2 family members and their BH3-only interactors plays an important role in controlling tissue homeostasis and deregulation can lead to cancer development. An increased understanding of the alterations of Bcl-2 members and their network of interactions in cancer could be useful to better exploit them as therapeutic targets. Through this study, we illustrate a computational workflow aimed at providing insight into: (i) the expression of pro-survival Bcl-2 members and their interactors in a certain cancer type, (ii) elucidating the functional interaction between them, and predicting the effects of substitutions on these interactions, and (iii) identifying alterations which could

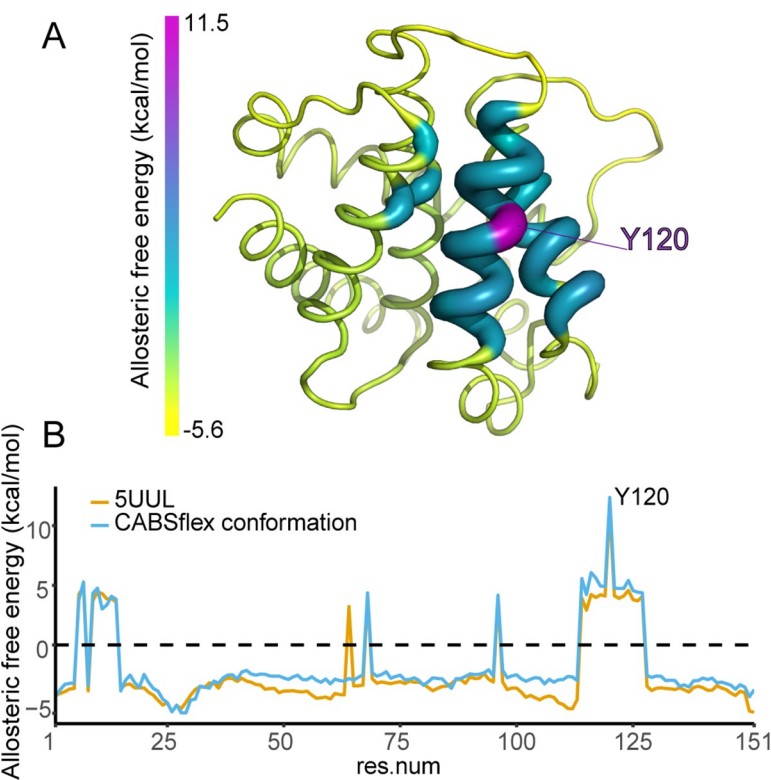

**Fig 7. Y120C as an allosteric mutation.** Using the approach implemented by *AlloSigMA*, we predict that the mutation of Y120 with a smaller residue (i.e., cys) could have a destabilizing impact on the network of contacts in Bcl2a1, altering the allosteric communication with the BH3-binding interface. **A)** We report the per-residue variations of allosteric free energy upon the Y120C mutation on the X-ray structure of Bcl2a1 in complex with the Puma BH3 (PDB entry 5UUL, as a cartoon) using a shared of colors from yellow (negative values) to purple (positive values). **B)** We compare the allosteric free energies calculated for the X-ray structure of Bcl2a1 with Puma BH3 with the ones calculated using a structure from the *CABS-Flex*, in which the sidechain of Y120 and the ones of its surroundings featured the largest deviation from the starting structure, to account for the ensemble heterogeneity.

impact on the turnover of the protein, altering its structural stability. As an example, we applied this workflow on the BRCA data from The Cancer Genome Atlas (TCGA). Our framework can be extended to any other cancer datasets deposited in the NCI Genomic Data Commons (GDC) or data from similar genomic initiatives.

Despite the importance of the Bcl-2 family conserved BH3 motif in mediating protein-protein interactions, a BH3 consensus motif is elusive [15]. We defined the motif for our search of candidate interactors in light of literature reports and a recent work in which the BH3 motif has been redefined as a short linear motif [16,42,43]. We allowed a certain degree of flexibility in certain conserved positions to have a broader coverage and prevent the removal, in our search, of possible non-conventional BH3-like proteins. The motif was applied to filter interaction partners of the Bcl-2 family members, extracted from an integrated curation of protein-protein interactions [47], providing a collection of more than 250 possible BH3-containing proteins of which 26 have been experimentally validated in literature. The remaining candidates could be interesting targets for experimental validation upon verification that they are in disordered regions or exposed helical regions of the corresponding proteins, a requirement for a BH3-like region.

The pro-survival members were shown to be up-regulated in a variety of tumor types [20,21] and were considered to contribute to tumorigenesis and therapeutic resistance [24,25]. We notice that, despite the classification of Bcl-2 members as either inhibitor or executioner of

apoptosis, their regulation in cancers is far from black and white, and the regulation is to a high degree tissue-context-dependent. For example, one could expect that pro-survival BCL-2 gene levels would largely be up-regulated in cancer types, but it has been demonstrated that such trend cannot be expected to be ubiquitous in cancers [21]. Further studies into the alterations of the pro-survival members at the mRNA and protein level in specific cancer (sub) types are necessary to generate the knowledge required to guide and optimize anti-cancer treatments. This point is especially critical since the same Bcl-2 proteins and BH3-containing interactors are not necessarily the fundamental ones to target for all the cancer types. Moreover, Bcl-2 proteins can compensate for each others loss and contribute to resistance to BH3 mimetics [7]. Here, from the analysis of the TCGA-BRCA dataset, we uncovered the gene expression landscape of globular Bcl-2 members and their putative BH3-like interactors in breast cancer. We found a marked signature for the pro-survival BCL2A1 gene, which is up-regulated in breast cancer and its subtypes. The expression and function of pro-survival BC2LA1 in normal tissues appears to be linked to the immune system in which, it seems that development of inflammasomes increases the expression of BCL2A1, consequently protecting pro-inflammatory cells from apoptosis [75]. Moreover, Bcl2a1 has a physiological function in the mammary glands, where its overexpression has been linked to the prevention of mammary gland involution by apoptosis [76]. A study, where different solid tumor tissues were compared, found the highest expression of BCL2A1 in breast cancers [21]. Another study, comparing expression levels between stages of breast cancer, found an association with a worse survival of the patient and high expression levels of BCL2A1 [77]. Apart from the likely role in tumorigenesis, Bcl2a1 induces chemotherapeutic resistance by suppressing apoptosis upon toxic stimuli, consequently preventing cell death. Overexpression of BCL2A1 in cell lines has been found to promote resistance to different cancer drugs including the BH3 mimetic ABT-737, a specific inhibitor targeting Bcl-2, Bcl-xL, and Bcl-w [29]. Due to their pivotal role as inhibitors of apoptosis, pro-survival Bcl-2 proteins have been considered promising targets for anti-cancer therapy. Progress has been made and several BH3 mimetics, which can target the hydrophobic cleft in pro-survival members, have been developed with promising perspectives [78–80]. These mimetics have been successful in inhibiting Bcl-2, Bcl-xL, and Bcl-w, but not Bcl2a1, or they have been broad-spectrum inhibitors with differing affinities depending on the pro-survival target proteins. In spite of the general progress, no potent and selective BH3 mimetics, targeting Bcl2a1 has been demonstrated so far [26]. Unraveling to what extent putative BH3-like interactors are deregulated in breast cancer and clarifying their possible interaction with Bcl2a1 at the structural level, might provide a valuable source of information. Identifying possible Bcl2a1 selective interactors could serve as templates for the design of BH3 mimetics, targeting and preventing its pro-survival role in tumors. An interesting approach to drug design for this protein has been proposed and can benefit from further knowledge on the interactome and specificity towards this underappreciated Bcl-2 family member [63].

We here find two putative BH3-containing interactors of interest in the context of Bcl2a1, i.e., Hrk and Nr4a1. These interactors are down-regulated in the TCGA-BRCA dataset and in different BRCA subtypes, accompanied by up-regulation of Bcl2a1, suggesting a signature of cell death evasion, which is not compensated by changes in the pro-apoptotic Bcl-2 family members. Another interesting predicted BH3-only interactor is Slc9A3r1 in the BRCA Basal subtype. Hrk has already been reported as a BH3-containing protein and its interaction with other Bcl-2 family members has been addressed experimentally [52,60]. Our study suggests that more extensive investigations into the interaction and cellular role of Bcl2a1 are needed. On the other hand, Nr4a1 would also need to be studied as a possible new and non-canonical BH3-containing protein.

We provided a model of interaction for Bcl2a1 and the two BH3-like motifs of Hrk, together with a deep mutational scanning, allowing for the identification of the possible molecular

determinants of their binding mode. One of the two BH3 motifs that we predicted for Hrk might be located in a transmembrane region [60]. We speculate that this could act as a "conditional" BH3, which might act as a sensor for structural changes induced by cellular conditions that can dissociate Hrk from the membrane and subsequently bind to Bcl-2 family members in the BH3-binding groove. Experiments addressing this hypothesis could shed new light on the Hrk mechanism of action and rule out that the motif predicted by our study is not a false positive.

A proper assessment of the impact of mutations on both the stability of Bcl-2 family members and of the binding affinity to BH3 interactors, is critical to understand the functional capacity of pro-survival proteins to propagate apoptosis. Drug resistance in anti-cancer treatments continues to be one of the leading reasons for unsuccessful treatments and several studies have linked mutations in Bcl-2 family members to altered sensitivity or resistance to BH3-mimetics [13,36,37]. In general, a structure-based functional and stability assessment of mutational data have lagged behind the growth of data generated from modern high-throughput techniques. Here, we applied a high-throughput computational mutational scan to predict the effects of missense mutations found in breast cancer samples, on both the structural stability of Bcl2a1, and the binding between Bcl2a1 and BH3-only proteins. This high-throughput approach additionally permitted us to evaluate the general effects of any amino-acid substitution on stability and binding. Moreover, it allowed us to suggest important positions in the BH3 region or in the Bcl2a1 protein for their interaction. For example, we shed light on the requirement for small side chain residues in proximity of the hydrophobic residues for interaction with h1 and h3 hydrophobic pockets of Bcl2a1, along with the possibility to replace the conserved negatively charged residue of the BH3 motif with the cognate polar residues, i.e., asparagine and glutamine. Moreover, we show how threonine occupying one of the hydrophobic pockets might be suboptimal for binding, knowledge which could be exploited for the design of higher affinity binders.

The deep mutational scanning allowed us to provide a more comprehensive view, beyond mere changes in expression levels, of the alterations of Bcl2a1 in breast cancer. In particular, we predict three mutations with different effects on the protein (i.e., L9R, M75R and Y120C). We also estimated the occurrence of these mutations across tumor samples in general, not only in breast cancer. In particular, we analyzed other TCGA datasets and the International Cancer Genome Consortium (see GitHub repository associated with the publication for more details) and we only found these mutations of BCL2A1 associated with breast cancer samples.

The Bcl2a1 mutations in breast cancer are not predicted to locally change the binding affinity with the BH3 only proteins used in our study. Two variants (i.e., L99R and M75R) are predicted with a marked impact on protein stability, suggesting that despite BCL2A1 up-regulation, the corresponding protein variant could be compromised due to increased turnover in some of the samples. This result clearly demonstrated how important it is to account for the compensatory effects that can be produced by different layers of modification occurring at the same time in a sample when analyzing cancer alterations associated with a certain group of genes. Finally, using a Protein Structure Network approach, we identify a mutation site (Y120), which might trigger allosteric effects to the BH3 binding groove and, as such, could be a long-range modulator of the Bcl2a1 protein.

## Conclusions

In summary, we here propose an integration of bioinformatics approaches, linking -omics data to structural ensembles, to unveil the pro-survival Bcl-2 signature in cancer. We provide a computational workflow to uncover the gene expression landscape of the complex protein-protein interaction network for the regulation of Bcl-2 family members, to analyze the structures of

these complexes and the impact of mutations. Moreover, we used a high-throughput *in silico* mutagenesis approach to identify functionally important residues in the pro-survival members and their interactors. Our study allowed us: i) to predict new BH3-only targets for future validation; ii) to revise the role of the invariant salt-bridge for interaction between Bcl-2 and BH3-only proteins; iii) to propose the complexes between Bcl2a1 and Hrk or Nr4a1, as new potential targets in breast cancer and iv) to identify three damaging mutations of Bcl2a1 for protein stability (L99R and M75R) or with allosteric effects (Y120C). Our study highlights the prospects of an integrative bioinformatics approach for the identification of new targets for BH3 mimetics. For example, we could apply these methods to identify substitutions in pro-survival BH3-only interactors that would reduce binding to other pro-survival members without substantially weakening the binding to the selected target. Lastly, we note that the approach, here applied to the study of pro-survival proteins, could be extended to anti-apoptotic members. For example, the assessment of cancer mutations to classify damaging or neutral mutations, would also be relevant with a focus on the anti-apoptotic members of the Bcl-2 family.

## Materials and methods

To reproduce this study, we released a GitHub repository where data, scripts, and guidelines are deposited (https://github.com/ELELAB/bcl_bh3only_breast_cancer).

### Identification of BH3 motif containing interaction partners

We used the *Integrated Interactions Database* (*IID*) [81] of tissue and organism-specific interactions downloaded on February 6th, 2018 (version 2017–04), to retrieve known interactions partners of the globular Bcl-2 family members (Uniprot identifiers: Bcl-2; P10415, Bcl-xL; Q07817, Bcl-w; Q92843, Mcl-1; Q07820, Bcl2-l10; Q9HD36, Bcl2a1; Q16548, Bok; Q9UMX3, Bax; Q07812, Bak; Q16611, Bcl2l12; Q9HB09, Bcl2l13; Q9BXK5, Bcl2l14; Q9BZR8, and Bcl2l15; Q5TBC7) in human tissues. Subsequently, we filtered the interaction partners to retain only those containing the definition of a consensus BH3 motif described in the results. The protein-protein interactions were visualized and analyzed as a network using *Cytoscape* [82]. Upon consultation of the recently released *BCL2DB* database [83], we discovered that some of the Bcl-2 family members are better classified as Bid-like proteins (i.e., Bcl2l12-15) and we discarded them from the analyses.

### Analysis of TCGA-BRCA RNA-seq data

For this study, we aggregated RNA-Seq BRCA data from The Cancer Genome Atlas (TCGA), using *TCGAbiolinks version 2.7.21* [84,85]. The data are accessible through the NCI Genomic Data Commons (GDC) data portal (https://portal.gdc.cancer.gov). The GDC Data Portal provides access to the subset of TCGA data that have been harmonized (i.e., HTseq read mapping) against GRCh38 (hg38).

The aggregated data were pre-processed, normalized, and filtered prior to analysis, using different *TCGAbiolinks* functions. We pre-processed the data using the function *TCGAanalyze Preprocessing*, estimating the Spearman correlation coefficient among all samples. Samples with a correlation lower than 0.6 were identified as possible outliers and removed. It has been demonstrated that divergent tumor purity levels can lead to a false interpretation of differentially expressed genes between cancer and normal samples, as it may induce a confounding effect in the analysis of transcriptomic dataset [86]. To account for this possible effect, we filtered samples according to a derived consensus measurement of purity of 0.6 [86] as implemented in *TCGAbiolinks* [85]. We normalized the data to adjusts for external factors that were not of biological interest and to ensure that expression distributions of each sample were similar across the data. We applied the function *TCGAanalyze Normalization*, implementing (i)

within-lane normalization to adjust for GC-content effect on read counts [87] and (ii) between-lane normalization to adjust for distributional differences between lanes, i.e., sequencing depth [88]. Lastly, the data were full quantile filtered, using a threshold of 0.25, implemented in the function *TCGAanalyze Filtering* to remove features with low expression across the samples. We retained only samples containing the PAM50 intrinsic molecular subtypes, along with protein coding genes.

To explore the global structure of the high-dimensional dataset, we applied Principal Component Analysis (PCA) with the aim of (i) examining to what extent differential expression within the primary conditions of interest, could be distinguished, along with (ii) identifying possible batch effects. PCA was computed using the *prcomp* function from the R package *stats*. The exploratory analyses were undertaken on normalized log2 transformed read counts to relieve the heteroscedastic behavior of raw read counts. A pseudo-count of 1 was added to avoid taking the log of zero. We performed differential expression analysis using the Bioconductor package *limma* [89]. *Limma* integrates a range of statistical methods for effective analysis of gene expression experiments. At its core lies the ability to fit gene-wise (rows) linear models to the matrix of expression levels. This approach allows for flexibility in the sense that entire experiments as an integrated whole, can be analyzed, rather than step-by-step comparisons between pairs of treatments. Gene-wise linear models empower the sharing of information between samples, allowing one to model correlations that might be present between samples due to repeated measures or the presence of covariates. As of such, linear models allow for the adjustment of effects of multiple experimental factors or batch effects. The linear models describe how the coefficients (treatments) are assigned to different samples. Another important statistical component of *limma* is the empirical Bayes procedure, which facilitates the moderation of the gene-wise variances. This method estimates an optimal variance for each gene as a trade-off between the gene-wise variance, procured for that gene alone, and the global variance across all genes. *limma* linear modeling is conducted on log-CPM values, assumed to be approximately normally distributed and with an independent mean-variance relationship. It has been demonstrated that for RNA-Seq and other sequence count data, the variance is often dependent of the mean [90]. To remove heteroscedasticity, we applied the *voom* function, converting the mean-variance relationship through lowess fit and subsequently uses this to estimate gene-wise variances. For each gene, the inverse of the variance is then applied as "precision weight" in the downstream *limma* framework [48]. We adjusted for multiple testing using the Benjamini & Hochberg procedure of controlling the false discovery rate (FDR) or adjusted p-value. Significance was defined using an adjusted p-value cutoff of 0.05 together with a log-fold-change (logFC) threshold of 1 or -1 (for up- and down-regulated genes, respectively). Differentially expressed genes were visualized in a volcano plot, created using the *TCGAVisualize volcano* function of *TCGAbiolinks*. We included directly in the design matrix the information on the Tissue Source Site (TSS), upon exploration with PCA. We did not incorporate the effects of plate as we could see from the data that the plates of interest were from the same TSS. Thus, the TSS was treated as a surrogate to avoid adding and extra parameter and associated degrees of freedom.

## Modeling of protein-peptide complexes

We modeled protein-peptide interactions with the scope of: (i) predicting the binding interface and the 3D structure of the complex between Bcl2a1 and Hrk BH3 regions, and (ii) identifying the location of the cancer mutations of Bcl2a1.

To model protein-peptide interactions, we applied comparative modeling, implemented in the program *MODELLER* v.9.15 [91], generating ten models for each alignment. *MODELLER*

carries out comparative protein structure modeling by satisfying spatial protein structure restraints and optimizing the structure until a model that best satisfies the spatial restraints is acquired. In our modeling, we used as additional restraints the distance between V74 of the hydrophobic cleft of the template Bcl2a1 (chain A) and the invariant leucine for *h2* in each of the target BH3 peptides (chain B). To infer reliability and discriminate between models calculated from the same alignment, we applied statistically optimized atomic potentials, specially trained for scoring and assessing protein-peptide interaction [92]. We used the web server *VADAR* v.1.8 [93] to further assess the quality of the models. One model for each alignment was retained after these assessments. As a template structure, we used the known X-ray 3D structure of Bcl2a1 in complex with the BH3 peptide from the canonical BH3-only protein Puma (Bcl2-binding component 3, PDB ID 5UUL, R = 1.33 Å [59]). We generated the models of the complexes between Bcl2a1 and two BH3-like peptides or Hrk (Hrk_1, residues 28–50 and Hrk_2, residues 63–85).

## Identification of mutations reported in cancer genomics datasets

We retrieved known missense mutations in the coding regions of BCL2A1 and HRK using the *MuTect2* pipeline [94] for the TCGA-BRCA samples, which compares tumor to a pool of normal samples to find somatic variations. We used the pipeline as implemented in the *TCGAbiolinks* function *GDCquery_Maf*. We integrated this search with breast cancer mutations deposited in other studies available in *CBioPortal* [95] and *COSMIC* [96]. We also verified that the mutations of interest were not found in *ExAC* [97] as polymorphisms, which occur at high frequency in the health population.

## Structure-based prediction of the functional impact of mutations

We used the *FoldX* (http://foldxsuite.crg.eu) empirical force field to predict changes in stability and interaction energies [98]. The *FoldX* energy function is obtained using a union of physical energy terms (e.g., van der Waals interactions, hydrogen bonding, electrostatics, and solvation), statistical energy terms, and structural descriptors that have been found important for protein stability. We used an in-house *Python* wrapper, *MutateX* [99] to support the systematic substitution of all wild-type residues to any of the 20 canonical amino acids, as recently applied to other cases of study [69,70,100]. With this tool we conducted *in silico* saturation mutagenesis, predicting ΔΔG values for all possible mutations in our modeled complexes. We applied the *RepairPDB* module from *FoldX,* optimizing the conformation of the model by repairing residues characterized by unfavorable torsion angles or, Van der Waals clashes. Subsequently, mutagenesis was carried out, applying the *BuildModel* module from *FoldX,* independently mutating each residue at every position and calculating the ΔΔG values. The prediction error of *FoldX* lies around 0.8 kcal/mol [98]. To infer the reliability of the predictions and discriminate between neutral and deleterious mutations, we applied a threshold of 1.6 kcal/mol (i.e., twice the prediction error). For visualization purposes, we applied a ΔΔG cutoff of 5 kcal/mol when plotting results of the deep mutational scan. The cutoff was derived by investigating the distribution of experimental ΔΔG values from the *ProTherm* database [101]. The vast majority of experimental ΔΔG values fall within -2.5 and 5 kcal/mol and, as such, *FoldX* predicted substitutions exceeding this value might be overestimated. Other details on the saturation mutagenesis protocol are provided in ref.[99]

## Protein structure network analysis

We used the *PyInteraph* suite [65] to derive a contact-based Protein Structure Network (PSN) [66] for the complex between Bcl2a1 and the BH3 peptides. We used, the structure of the

complex of BCl2a1 with Puma, which has been solved by X-ray (PDB entry 5UUL)). Since the *PyInteraph* method has been designed to work on a structural ensemble, we collected a representative ensemble of ten conformations for this complex using *CABS_Flex 2.0* [68].

We considered any two residues whose side-chain centers of mass lied within 5.0 Å as interaction pairs in the PSN. This cut-off was selected as suggested by a recent benchmarking of the method [66]. We also applied a 20% cutoff to the persistence of the interaction to filter out transient and spurious interactions in the PSN, as previously suggested [65,102]. We included all the residues with the exception of glycine for the contact analysis. We applied a variant of the depth-first search algorithm to identify the shortest paths of communication, whereas hubs were defined as residues with a degree higher than three (i.e., linked by more than three edges in the network), as generally applied to PSNs [103].

## Supporting information

**S1 Table. The table contains the full list of interactions for each Bcl-2 protein as retrieved from the *IID* database.**
(XLSX)

**S2 Table. The table contains the information on the literature-based curation of BH3-containing proteins among the Bcl-2 interactors for which a BH3 motif has been identified by our motif search.**
(XLSX)

**S3 Table. We here report the results of the BH3 motifs found using regular expression for each of the Bcl-2 interactors.**
(TXT)

**S4 Table. We here report the full list of results of the differential expression analyses for each comparison and gene under investigation in our study.**
(XLSX)

**S5 Table. We here report the results from the analyses of missense mutations in BC2LA1 using the *Mutect2* pipeline applied to the TCGA BRCA dataset.**
(XLSX)

**S6 Table. We here report the results from the deep mutational scan with *Foldx* of the free state of Bcl2a1.** The other mutational scans are available in the GitHub repository associated with the publication.
(XLSX)

**S7 Table. We here report the per-residue *ConSurf* results.**
(TXT)

**S1 Fig. We here report the heatmap from the deep mutational scan with *FoldX* to estimate the binding free energy for the Bcl2a1-Puma complex.**
(PDF)

## Acknowledgments

The results shown here are in part based upon data generated by the TCGA Research Network: https://www.cancer.gov/tcga. The calculations described in this paper were performed using the DeiC National Life Science Supercomputer Computerome at DTU (Denmark).

## Author Contributions

**Conceptualization:** Elena Papaleo.

**Data curation:** Simon Mathis Kønig, Vendela Rissler, Elena Papaleo.

**Formal analysis:** Simon Mathis Kønig, Vendela Rissler, Matteo Lambrughi, Elena Papaleo.

**Funding acquisition:** Elena Papaleo.

**Investigation:** Simon Mathis Kønig, Elena Papaleo.

**Methodology:** Simon Mathis Kønig, Vendela Rissler, Thilde Terkelsen, Matteo Lambrughi, Elena Papaleo.

**Project administration:** Elena Papaleo.

**Resources:** Elena Papaleo.

**Supervision:** Elena Papaleo.

**Visualization:** Simon Mathis Kønig, Elena Papaleo.

**Writing – original draft:** Simon Mathis Kønig, Elena Papaleo.

**Writing – review & editing:** Vendela Rissler, Thilde Terkelsen, Matteo Lambrughi, Elena Papaleo.

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
