## [Decision Letter · Decision Letter 0]

17 Aug 2019

Dear Dr Papaleo,

Thank you very much for submitting your manuscript, 'Alterations of the pro-survival Bcl-2 protein interactome in breast cancer at the transcriptional, mutational and structural level', to PLOS Computational Biology. As with all papers submitted to the journal, yours was fully evaluated by the PLOS Computational Biology editorial team, and in this case, by independent peer reviewers. The reviewers appreciated the attention to an important topic but identified some aspects of the manuscript that should be improved.

We would therefore like to ask you to modify the manuscript according to the review recommendations before we can consider your manuscript for acceptance. Your revisions should address the specific points made by each reviewer and we encourage you to respond to particular issues Please note while forming your response, if your article is accepted, you may have the opportunity to make the peer review history publicly available. The record will include editor decision letters (with reviews) and your responses to reviewer comments. If eligible, we will contact you to opt in or out.raised.

- Supporting Information uploaded as separate files, titled 'Dataset', 'Figure', 'Table', 'Text', 'Protocol', 'Audio', or 'Video'.

We hope to receive your revised manuscript within the next 30 days. If you anticipate any delay in its return, we ask that you let us know the expected resubmission date by email at ploscompbiol@plos.org.

Sincerely,

Igor N Berezovsky

Guest Editor

PLOS Computational Biology

Ruth Nussinov

Editor-in-Chief

PLOS Computational Biology

[LINK]

Reviewer's Responses to Questions

**Comments to the Authors:**

Reviewer #1: In this paper, the authors obtained an interactome for pro-survival Bcl-2 proteins and analyzed the changes in gene expression of the proteins and their putative BH3-only interactors. The paper highlighted two genes encoding BH3-only proteins that are downregulated along with up-regulated BCL2-A1 in some comparisons, and residues that are important in binding BH3-only proteins, suggesting their potential in guiding the design of BH3 mimetics.

Page 5, line 223. One would expect that most pro-survival BCL-2 genes would be up-regulated in cancer samples, however this is not what is observed, except in BCL2-A1. Can the authors explain this?

Page 10, line 455. How are “collisional events” defined and measured?

Page 10, line 456 and page 12, 570. The authors suggested that the Y120C on the surface may be an allosteric mutation, based on the amino acid networks that connect the residue to the probes. To obtain a direct estimation of the allosteric effect in the probes upon the mutation, the authors could use some computational tools available online. Also, other possible scenarios are aberrant protein-protein interaction or disrupted PTMs caused by Y120C mutation, leading to cancer. Is the residue known to be involved in PPI or PTM?

Figure 6, it is not clear where Y120 is located in the structure, please label the residues. If I understand correctly, the empty circles in (d) represent the presumably overestimated mean ∆∆g from FoldX, if that is the case, it may be better to omit values above 5 kcal/mol in the figure. Also, it may be of interest to some to show and label the residues with significant negative values from (d).

Minor points:

Pg 3 line 110, a word is missing after “… two independent”

Pg 4, line 195, please change BH3-containin to containing

Pg 6, line 259, please remove “and” before “were differentially”

Pg 6, line 297, please remove ‘s’ after available

Table 2, please replace the comma in row 3, column 7

Reviewer #2: Mathis et al report a comprehensive study to assess the impact of missense mutations of Bcl2a1 which could either contribute to its deregulation or partially compensate for its up-regulation. The paper is well written and the overall approach is described in detail. Results are interesting and deserve further experimental validations to study the predicted impact of somatic mutations of Bcl2a1.

I suggest to better clarify what the author call "multiscale" bioinformatic approach, for adding an overall picture of the process that they adopt. I would also suggest to quantify the incidence of the mutations L99R, M75R and Y120C across tumors, for example using the TCGA or ICGC datasets. I would also recommend to add a section both in the abstract and in the conclusions with a summary of the main findings of the paper.

**Have all data underlying the figures and results presented in the manuscript been provided?**

Reviewer #1: None

Reviewer #2: Yes

PLOS authors have the option to publish the peer review history of their article (what does this mean?). If published, this will include your full peer review and any attached files.

Reviewer #1: No

Reviewer #2: No

---

## [Decision Letter · Decision Letter 1]

12 Oct 2019

Dear Dr Papaleo,

We are pleased to inform you that your manuscript 'Alterations of the pro-survival Bcl-2 protein interactome in breast cancer at the transcriptional, mutational and structural level' has been provisionally accepted for publication in PLOS Computational Biology.

In the meantime, please log into Editorial Manager at https://www.editorialmanager.com/pcompbiol/, click the "Update My Information" link at the top of the page, and update your user information to ensure an efficient production and billing process.

One of the goals of PLOS is to make science accessible to educators and the public. PLOS staff issue occasional press releases and make early versions of PLOS Computational Biology articles available to science writers and journalists. PLOS staff also collaborate with Communication and Public Information Offices and would be happy to work with the relevant people at your institution or funding agency. If your institution or funding agency is interested in promoting your findings, please ask them to coordinate their releases with PLOS (contact ploscompbiol@plos.org).

Thank you again for supporting Open Access publishing. We look forward to publishing your paper in PLOS Computational Biology.

Sincerely,

Igor N Berezovsky

Guest Editor

PLOS Computational Biology

Ruth Nussinov

Editor-in-Chief

PLOS Computational Biology

Reviewer's Responses to Questions

Comments to the Authors:

Please note here if the review is uploaded as an attachment.

Reviewer #1: All points have been addressed, the manuscript is suitable for publication.

Reviewer #2: My concerns were completely addressed

Have all data underlying the figures and results presented in the manuscript been provided?

Large-scale datasets should be made available via a public repository as described in the 

PLOS Computational Biology

data availability policy, and numerical data that underlies graphs or summary statistics should be provided in spreadsheet form as supporting information.

Reviewer #1: None

Reviewer #2: Yes

PLOS authors have the option to publish the peer review history of their article (what does this mean?). If published, this will include your full peer review and any attached files.

Do you want your identity to be public for this peer review?

 For information about this choice, including consent withdrawal, please see our Privacy Policy.

Reviewer #1: No

Reviewer #2: No

---

## [Editor Report · Acceptance letter]

25 Nov 2019

PCOMPBIOL-D-19-01138R1 

Alterations of the pro-survival Bcl-2 protein interactome in breast cancer at the transcriptional, mutational and structural level

Dear Dr Papaleo,

I am pleased to inform you that your manuscript has been formally accepted for publication in PLOS Computational Biology. Your manuscript is now with our production department and you will be notified of the publication date in due course.

With kind regards,

Matt Lyles
